# Can Administrative Health Data Improve the Gold Standard? Evidence from a Model of the Progression of Myocardial Infarction

**DOI:** 10.3390/ijerph18147385

**Published:** 2021-07-10

**Authors:** Son Nghiem, Jonathan Williams, Clifford Afoakwah, Quan Huynh, Shu-kay Ng, Joshua Byrnes

**Affiliations:** 1Centre for Applied Health Economics, School of Medicine and Dentistry, Griffith University, Brisbane, QLD 4111, Australia; c.afoakwah@griffith.edu.au (C.A.); s.ng@griffith.edu.au (S.-k.N.); j.byrnes@griffith.edu.au (J.B.); 2Department of Statistics, North Carolina State University, Raleigh, NC 27695, USA; jwilli27@ncsu.edu; 3Baker Heart and Diabetes Institute, Melbourne, VIC 3004, Australia; quan.huynh@baker.edu.au

**Keywords:** administrative data, myocardial infarction, disease progression, Australia

## Abstract

*Background*: Myocardial infarction (MI), remains one of the leading causes of death and disability globally but publications on the progression of MI using data from the real world are limited. Multistate models have been widely used to estimate transition rates between disease states to evaluate the cost-effectiveness of healthcare interventions. We apply a Bayesian multistate hidden Markov model to investigate the progression of MI using a longitudinal dataset from Queensland, Australia. *Objective*: To apply a new model to investigate the progression of myocardial infarction (MI) and to show the potential to use administrative data for economic evaluation and modeling disease progression. *Methods*: The cohort includes 135,399 patients admitted to public hospitals in Queensland, Australia, in 2010 treatment of cardiovascular diseases. Any subsequent hospitalizations of these patients were followed until 2015. This study focused on the sub-cohort of 8705 patients hospitalized for MI. We apply a Bayesian multistate hidden Markov model to estimate transition rates between health states of MI patients and adjust for delayed enrolment biases and misclassification errors. We also estimate the association between age, sex, and ethnicity with the progression of MI. *Results*: On average, the risk of developing Non-ST segment elevation myocardial infarction (NSTEMI) was 8.7%, and ST-segment elevation myocardial infarction (STEMI) was 4.3%. The risk varied with age, sex, and ethnicity. The progression rates to STEMI or NSTEMI were higher among males, Indigenous, or elderly patients. For example, the risk of STEMI among males was 4.35%, while the corresponding figure for females was 3.71%. After adjustment for misclassification, the probability of STEMI increased by 1.2%, while NSTEMI increased by 1.4%. *Conclusions*: This study shows that administrative health data were useful to estimate factors determining the risk of MI and the progression of this health condition. It also shows that misclassification may cause the incidence of MI to be under-estimated.

## 1. Introduction

Myocardial infarction (MI), popularly known as a heart attack, is typically caused by obstruction of blood flow either by plaques in the coronary arteries [1] or, less frequently, by other obstructing mechanisms, such as spasm of coronary arteries [2]. Although the incidence of MI is declining in some developed countries [3], Australia is an outlier with a very high, and increasing, incidence rate [4,5]. A better understanding of the progression of an MI can lead to the development of interventions to reduce the incidence and improve health outcomes.

Randomized control trials remain the “gold standard” [6] in evaluating healthcare interventions and modeling the progression of diseases under systematic interventions. Despite their popularity, randomized control trials have drawbacks. First, the sample size is relatively small, ranging from dozens to hundreds [7]. Second, they are expensive and time-consuming [8,9]. Third, there is a big gap between experiment and practice [8,10]. In contrast, administrative health data do not have these limitations because they capture the whole population, and are often readily available and always reflect the real world [11].

Multistate survival models [12] are long-established and well-accepted tools to investigate the progression of diseases. The main assumption in most multistate models is that health states are observed with certainty, which is not always true. Fortunately, we can also use a hidden Markov model, which relaxes this assumption, to consider misclassifications in identifying health states [13].

In this paper, we apply a hidden Markov model [14] to investigate the progression of MI in a cohort of linkage hospital admission data from Queensland, Australia. We show that the risk of MI is under-estimated if misclassification is ignored. The bias-corrected transition rates estimated in this study can be used in economic evaluations of health interventions. Moreover, we believe that the use of administrative data to investigate the progression of diseases, as we have done here, will increase, and administrative data can improve the “gold standard” in the future.

## 2. Methods

### 2.1. Data Sources

The main data used in this study were the Queensland cardiac linkage cohort of all cardiovascular diseases (CVD)-related hospitalizations in 2010. All later admissions of those patients were followed until the end of 2015. The data related to these admissions were linked and collected from the Queensland Hospital Admitted Patient Data Collection (QHAPDC), Emergency Department Data Collection (EDDC), Registrar of General Deaths (RGD), and the National Hospital Cost Data Collection (NHCDC). 

We used the International Classification of Diseases—10th Revision codes of primary diagnosis to classify MI. The two most common types of MI include Non-ST-segment Elevation Myocardial Infarction (NSTEMI, and STEMI [15]. NSTEMI was identified using the code I21.4, while STEMI was identified using codes I21.0, I21.1, I21.2, or I21.3. Unspecified and other forms of MI (codes I21.9 and I21.A) were less common and not clearly identified and hence excluded from the analysis [16].

### 2.2. Ethics Committee Approval

Data were released in accordance with the research provisions of the Queensland Public Health Act 2005 [17]. Data collection and analyses in this study were conducted according to the Declaration of Helsinki [18]. Ethical approval was obtained from the Griffith University and Queensland Health Human Research Ethics Committees (Ref No: 2017/001). The data were stored and accessed using the Secure Unified Research Environment [19]. 

### 2.3. Study Population

The original dataset included 135,399 individuals with 1.8 million admissions during the study period. We excluded 59,074 individuals who were not admitted for the first time to control for left-censoring bias. This left-censoring bias could be upwards as those admitted to hospitals before 2010 could be at a more severe stage of disease progression. However, the left-censoring bias could also be downward because those who first hospitalized before 2010 and still alive in 2010 were fitter (i.e., the survival of the fittest). 

Among the remaining individuals, 30,494 were excluded because the cardiovascular disease was not listed as a primary diagnosis code (i.e., CVD was not the primary cause of the index admission). These patients were hospitalized primarily for treatments of other health conditions (e.g., kidney failure); and presented with CVDs as subsequent diagnoses, or as primary diagnoses in subsequent admissions. Finally, we focused on patients who had at least one episode of MI during the study period. The final sample included 8705 patients with 70,205 admissions from January 2010 to December 2015.

### 2.4. Hidden Markov Model and the Transition Rate Matrix

We apply a multistate hidden Markov model [13,14] to estimate the progression of MI. As the name suggests, a multistate Markov model refers to a Markov process (e.g., over health states), where a future state depends solely on the current state. In a standard Markov model, it is assumed that states were defined without errors. The hidden Markov model relaxes this assumption and allows misclassification, which can be due to measurement errors, to occur. 

In this study, we specified the progression of MI using four health states: the initial state (i.e., well); two transitory states (i.e., NSTEMI and STEMI); and the absorbing state (i.e., dead). Note that the ‘well’ state was defined as the absence of MI, and, hence, it does not necessarily indicate good health as patients may have been hospitalized for other cardiac conditions (e.g., angina).

The matrix of transition rates from one health state to another can be estimated using the exponential matrix $P\left(t\right)=e^{tQ}$, where *Q* is the transition rate matrix [13,14]. Each component of the *Q* matrix represents changes in transition rates by one unit of time. Components of the transition rate matrix can be affected by covariates such as age, sex, and Indigenous status: $\log\left(q_{l}\right)=\beta_{0}^{l}+\beta_{1}^{l}\cdot Age+\beta_{2}^{l}\cdot Sex+\beta_{3}^{l}.$ *Ethnicity*, where $q_{l}$ (l = 1,2…6) were elements of the transition rate matrix (Q).
$$Q=\left[\begin{array}{cccc} -q_{1}-q_{2}-q_{3} & q_{1} & q_{2} & q_{3}\\ 0 & -q_{4}-q_{5} & q_{4} & q_{5}\\ 0 & 0 & -q_{6} & q_{6}\\ 0 & 0 & 0 & 0 \end{array}\right]$$

### 2.5. Statistical Analysis

With the exception of the absorbing ‘dead’ state, the remaining observed health states were manifested from their true states with potential misclassification [13]. The main source of misclassification is the unobserved nature of disease progression. For example, a person could have their artery blocked, the main cause of MI, long before the first hospitalization by MI. Another factor contributing to the potential misclassification in these data was the difficulty to clearly identify types of MI using the current ICD-10 [20]. For the convenience of computation, we define the progression of MI, including only forward transitions. This definition is in line with the objective of medical practices that healthcare interventions aim to reduce the incidence or improve health outcomes of patients with MI. 

Based on the literature [21] and the available data, we selected age, sex, and ethnicity as covariates of MI progression. Other risk factors (e.g., smoking status and body mass index) were not available in our dataset. The age variable was converted into integers such that transition rates remained homogenous within a year. We chose the log-linear functional form to link the selected covariates (age, sex, and ethnicity) to transition rates between health states. 

Death rate bias was a potential issue in this study. This bias arose when people who died of an MI prior to hospitalization were not included in admission records. Thus, the death rate among patients in hospital databases could be lower than that of the general population. Models unadjusted for death rate bias could lead to lower estimated death rates. We mitigated the death rate bias by selecting the overall population death rate of Australia [22] as a prior for the Bayesian estimator. 

The estimated progression of MI in this study could also be biased by delayed enrolment. This issue occurs when people who had their first MI at an older age (e.g., 70) are healthier than those who had the first MI at a younger age (e.g., 30). We address this delayed enrolment bias issue by estimating the transition rates from the baseline health state at the first hospitalization. We also manually added censored observations for any year in the study period (2010–2015) where the patient had no admission prior to the current admission. For example, a patient hospitalized in 2010 with angina returned to a hospital in 2013 with an NSTEMI. In this case, two censored rows were added to model the potential that the patient could develop an NSTEMI in 2011 or 2012. After adding censored rows, the number of observations increased from 70,205 to 96,511. The 26,306 added observations come from 7251 people (83.3% of the sample) who had missing years between two adjacent admissions. 

The analysis was conducted in *R* [23] using a pre-released version of the *hmm* package [14].

## 3. Results

On average, the annual rates of NSTEMI and STEMI were 27.4% and 11.6%, respectively. The average mortality risk was 5.6%, while the average probability of having no heart attack (well) during the study period was 55.5%. 

The summary of health states by sex and ethnicity show that age is the most influential factor for transitioning between health states. The mortality risk increased from 1.1% for those aged less than 55 years to 11.7% for people aged 75 years and above (Table 1). The overwhelming effect of age could explain why the raw mortality risk did not differ by sex (*p*-value = 0.25) or ethnicity (*p*-value = 0.16). The age-standardized mortality rates show that males were more at risk than females, with a rate of 6% compared to 5.1% of females.

After age-standardizing, the mortality rate of Indigenous patients was 7%, while the rate for non-Indigenous patients was 5.5%. Males had a higher risk of STEMI (12.8% vs. 8.6%). Indigenous patients had the rate of having a STEMI lower by 1% compared to non-Indigenous patients. In addition, the proportion of Indigenous people in this sample (3.9%) was considerably higher than the Australian census estimate (2.8%) [24], suggesting a higher risk of MI among the Indigenous population. 

Overall, data reveal that age is the most influential factor affecting the health state of patients with MI. Differences in probabilities of the “well” state by age signals the potential of late enrolment bias may be present. Additionally, misclassification errors may exist as types of MI could not be clearly distinguished by ICD-10 codes [20]. We will mitigate both sources of bias in subsequent analysis. 

The estimated transition rate matrix at mean age confirmed with the data that NSTEMI was the most common at 8.46–8.70% (Table 2). STEMI was mostly developed from NSTEMI (2.4%). Males and Indigenous patients face a higher risk of developing a STEMI or NSTEMI, but the magnitude of differences was negligible.

The progression of MI also increased with age, but the effect sizes were negligible in most transitions. For example, each added year from the mean age of 67, the relative risk of having a STEMI increase by 1.5% [0.1,4.2%]. Thus, the relative risk of experiencing a STEMI is 60% lower for a 27-year-old individual than a 67-year-old (i.e., 40 years ×1.5%). However, because the level of transition rate was small, the difference in absolute risk of STEMI for this 40-year age gap was only around one percentage point.

Results of the misclassification prediction suggest that the prevalence of MI could be under-estimated (Table 3). Particularly, 3.7% of patients with an NSTEMI could be misclassified as “Well”, while 2.7% of patients in a STEMI could be misclassified as NSTEMI. Thus, the true risk of NSTEMI and STEMI should increase by 1.4% and 1.2%, respectively.

## 4. Discussion

### 4.1. Findings from Administrative Data Were Consistent with Randomized Trials

This study shows that administrative data can be used to investigate the progression of diseases, such as MI. Our findings are consistent with previous studies based on randomized trials [25], showing that males and Indigenous Australians have a higher mortality risk from an MI. However, we emphasize that both the absolute and relative measures of risk should be reported to reveal a true gap in MI between sexes and ethnic groups. In our study, the relative risk showed that males had a 63% greater risk of developing a STEMI despite the absolute difference being about 1%. Thus, reporting the sex and ethnicity gap using only relative or absolute difference may reach a distorted conclusion.

The estimated transition rates in this study provide several advantages for cost-effectiveness analyses of interventions for MI. Firstly, our population-based longitudinal dataset captures the whole population and so avoids problems associated with small sample sizes. Second, administrative data are readily available and hence less resource-intensive to collect. Third, administrative data can take into account factors that might not be controlled in randomized control trials. Overall, we argue that evidence from the real world (e.g., administrative data) should be used in medical decision making because of its obvious external validity and efficacy.

### 4.2. The Prevalence of MI Was Under-Estimated if Misclassification Is Ignored

Another important message from this study is that misclassification, if ignored, will result in an under-estimating risk of MI by 1.2–1.4%, which was substantial compared to the estimated prevalence of around 8.7% for non-severe and 4.3% for STEMI. This finding of substantial misclassification is consistent with previous findings in the USA [26,27] and Belgium [28]. However, previous studies attributed misclassification to errors in diagnosis, whereas misclassification in our study was attributable to the unobservable nature of the progression of diseases. Nevertheless, the finding of this study reaches the same conclusion that misclassification should be considered in modeling the progression of diseases [29].

Overall, administrative data have clear advantages over trial data in terms of large sample size, relatively cheaper to collect and the ability to consider differences between clinical trials and real-world medical practices. However, administrative data also have limitations with respect to no randomized control group and often lacks detailed information, for example, on lifestyle data. We also could not exclude a possibility of patient misclassification due to errors during ICD coding.

## 5. Conclusions

In conclusion, this study shows that administrative data provide an effective tool to model the progression of MI. Our study demonstrates that using administrative data in disease progression modeling will be more efficient and have stronger external validity. We also improve the reliability of the estimates by controlling for late enrolment bias and death rate bias. We found that the risk of MI could be under-estimated if misclassification is ignored. Ideally, a combination of evidence from randomized control trials evidence and administrative data should also be considered to take the best of both worlds. Our main results are consistent with the literature that age, sex, and ethnicity are the main drivers of heart attack risk.

## Figures and Tables

**Table 1 ijerph-18-07385-t001:** Proportions of patients in health states by sex, ethnicity, and age groups.

Characteristics	Health States			
	Well	NSTEMI	STEMI	Death
All cases	55.5%	27.4%	11.6%	5.6%
*By sex*				
Females	58.0%	27.6%	8.4%	6.0%
Males	54.1%	27.3%	13.3%	5.3%
*p*-value of the sex difference	<0.001	0.132	<0.001	0.254
*By ethnicity*				
Non-Indigenous	55.6%	27.4%	11.5%	5.6%
Indigenous	53.2%	29.3%	13.4%	4.1%
*p*-value of the ethnic difference	0.735	0.035	0.044	0.163
*By age groups*				
<55	56.3%	26.3%	16.2%	1.1%
55–64	57.5%	26.6%	13.9%	2.1%
64–74	56.8%	27.5%	10.9%	4.8%
75+	52.4%	28.6%	7.3%	11.7%
*p*-value of differences across age groups	<0.001	<0.001	<0.001	<0.001
*Age-standardized rates*				
Females	59.0%	27.3%	8.6%	5.1%
Males	53.8%	27.4%	12.8%	6.0%
Non-Indigenous	55.6%	27.3%	11.6%	5.5%
Indigenous	53.4%	29.0%	10.6%	7.0%

Note: Probabilities are normalized to make each row sum to one. *p*-values were obtained from χ^2^ tests.

**Table 2 ijerph-18-07385-t002:** Estimated transition rates by sex and ethnicity.

Transitions	Males	Females	Indigenous	Non-Indigenous
Well → NSTEMI	8.70%	8.46%	8.62%	8.61%
Well → STEMI	1.92%	1.29%	1.68%	1.69%
Well → Dead	2.83%	2.82%	2.81%	2.83%
NSTEMI → STEMI	2.43%	2.42%	2.41%	2.43%
NSTEMI → Dead	5.12%	5.08%	5.10%	5.11%
STEMI → Dead	4.68%	4.64%	4.68%	4.66%

**Table 3 ijerph-18-07385-t003:** Predicted misclassification probabilities.

Health States		Mean	95% Credible Set	
Observed = NSTEMI	True = Well	0.011	0.01	0.012
Observed = Well	True = NSTEMI	0.037	0.03	0.039
Observed = STEMI	True = NSTEMI	0.015	0.01	0.017
Observed = NSTEMI	True = STEMI	0.027	0.02	0.030

## Data Availability

Restrictions apply to the availability of these data. Data were obtained from Queensland Health and the Australian Institute of Health and Welfare and stored at the Secure Unified Research Environment (SURE) at the SAX Institute, Australia. Access to this data set is subject to ethical approval from all the data custodians.

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
