# Peer review of "Can Administrative Health Data Improve the Gold Standard? Evidence from a Model of the Progression of Myocardial Infarction"

_ijerph, 2021, doi:10.3390/ijerph18147385_

Round 1

Reviewer 1 Report

Dear Authors,

The manuscript brings contribution to science and I haven’t detected major concerns about its representation. 

Author Response

Thank you.

Reviewer 2 Report

Manuscript ID: ijerph-1283705

Manuscript title: Can Administrative Health Data Improve the Gold Standard? Evidence from a Model of the Progression of Myocardial Infarction

Major comments

  1. Methods, section 2.5 Statistical analysis. I agree with the interest in forward transitions. I believe backward transitions (with the obvious exclusion of death state) may be of interest to the clinicians as well. As these probabilities cannot be calculated from the current summaries, consider adding supplementary data with this analysis.

  1. Discussion. Most limitations of the study design and methods are disclosed along with the text. A summary of these limitations and the strengths of the study might improve the discussion.

  1. Conclusions. Consider rewriting this section as most statements concern the applied methods and not the presented results.

Minor comments

  1. Abstract (and other text passages). ‘the incidence of myocardial infarction under-reported’ or ‘under-estimated’?

  1. Introduction, 2nd paragraph. ‘Randomized control trials remain the “gold standard” [6] in evaluating healthcare interventions and modelling the progression of diseases’. Not sure RCT allows modelling the progression of diseases in general (also your reference [6] only mentions the first aim of a RCT). I suppose it can be said in particular ‘modelling the progression of diseases under systematic interventions.’

  1. Methods, section 2.4. Possible typo (term ‘Ethnicity’ is not in italic format as it is apparently outside the equation object).

  1. Methods, section 2.5 Statistical analysis. Report the statistical tests for the P-values in Table 1 (as reported in the table note).

  1. Results, 5th paragraph. ‘The estimated transition probability at mean age confirmed with the data that 181 NSTEMI was the most common at 8.6–8.7% (Table 2)’. Should 8.6 read 8.4 (8.46% in table 2)?

Author Response

Major comments

  1. Methods, section 2.5 Statistical analysis. I agree with the interest in forward transitions. I believe backward transitions (with the obvious exclusion of death state) may be of interest to the clinicians as well. As these probabilities cannot be calculated from the current summaries, consider adding supplementary data with this analysis.

Response

Thank you for the suggestion this is a limitation of this study that we will examine backward in future studies. It is a significant variation in the model and given the technical difficulty it is beyond this paper to include here.

  1. Discussion. Most limitations of the study design and methods are disclosed along with the text. A summary of these limitations and the strengths of the study might improve the discussion.

Response

We agree and following the reviewer’s comments have added a few sentences to summarise the limitations and strengths as follow.” Overall, administrative data have clear advantages over trial data in term of large sample size, relatively cheaper to collect and the ability to consider differences between clinical trials and real-world medical practices. However, administrative data also have limitations with respect to no randomised control group and often lacks detailed information, for example on lifestyle data. We also could not exclude a possibility of patient misclassification due to errors during ICD coding.”

  1. Conclusions. Consider rewriting this section as most statements concern the applied methods and not the presented results.

Response

Thank you for this suggestion, we agree and have added a summary of the results in the Conclusion section as follows “Our main results are consistent with the literature that age, sex and ethnicity are main drivers of heart attack risk.”

 Minor comments

  1. Abstract (and other text passages). ‘the incidence of myocardial infarction under-reported’ or ‘under-estimated’?

Response

Thanks for the interesting question. We believe that under-estimated is more relevant as under-reported may implicitly assume that the prevalence rate was reported lower on purpose. What we mean is that the current estimation method does not take into account the misclassification issue, which could result in lower prevalence rate. We have now used “under-estimated” in the whole paper.

  1. Introduction, 2nd paragraph. ‘Randomized control trials remain the “gold standard” [6] in evaluating healthcare interventions and modelling the progression of diseases’. Not sure RCT allows modelling the progression of diseases in general (also your reference [6] only mentions the first aim of a RCT). I suppose it can be said in particular ‘modelling the progression of diseases under systematic interventions.’

       Response

Thank you for the suggestion. We have added the suggested phrase ‘modelling the progression of diseases under systematic interventions.’

  1. Methods, section 2.4. Possible typo (term ‘Ethnicity’ is not in italic format as it is apparently outside the equation object).

Response

Thank you very much, it was a typo and is now fixed.

  1. Methods, section 2.5 Statistical analysis. Report the statistical tests for the P-values in Table 1 (as reported in the table note).

Response

We have included P-value in the last row of each results section.

  1. Results, 5th paragraph. ‘The estimated transition probability at mean age confirmed with the data that 181 NSTEMI was the most common at 8.6–8.7% (Table 2)’. Should 8.6 read 8.4 (8.46% in table 2)?

Response

Yes, the range should be from 8.46%-8.70%. This typo mistake has been corrected in the revised manuscript.

This manuscript is a resubmission of an earlier submission. The following is a list of the peer review reports and author responses from that submission.

Round 1

Reviewer 1 Report

This paper describes a study on myocardial infarction based on administrative data.

The paper is written sufficiently well. I just underline some points about this:

  • the sentence at line 182 should be revised "2.7% of patients in a STEMI could be misclassified as STEMI"
  • the title maybe could be changed into "Can Administrative Health Data Improve the Gold Standard? Evidence from a Model OF the Progression of Myocardial Infarction"
  • lines 97-98 need a reference
  • probabilities in Table 1 don't sum up to 1; this is evidenced in the footnote, but in my opinion, should be corrected
  • P-values about age in Table 1 are all 0?
  • captions of Tables 1 and 2 should be revised

On the other hand, the soundness and novelty of results are obviously very low, since the work just aims at obtaining results consistent with previous findings. Therefore, I would suggest to underline that the novelty stands in the approach adopted, revise the related work by examining previous similar approaches, and emphasize why it could be preferable (e.g., number of samples) but also which are its limits, with respect to the state of the art approaches (e.g., randomized trials).

Author Response

the sentence at line 182 should be revised "2.7% of patients in a STEMI could be misclassified as STEMI"

Response

Thank you, this is a typo. The correct one is “2.7% of patients in a STEMI could be misclassified as NSTEMI”

the title maybe could be changed into "Can Administrative Health Data Improve the Gold Standard? Evidence from a Model OF the Progression of Myocardial Infarction"

Response

We agree, and have inserted the missing “of” in the title.

lines 97-98 need a reference

Response

We added references 13,14.

probabilities in Table 1 don't sum up to 1; this is evidenced in the footnote, but in my opinion, should be corrected

Response

We have normalised the probabilities such that they sum up to one and have removed footnote.

P-values about age in Table 1 are all 0?

Response

P-values show as zeros when rounded to 3 decimal places but they’re not exactly zero. We have modified it to <0.001 as standard to report

captions of Tables 1 and 2 should be revised

Response

We have revised the captions as follow:

Table 1. Probabilities of health states by sex, ethnicity and age groups

Table 2. Transition probabilities by sex and ethnicity

On the other hand, the soundness and novelty of results are obviously very low, since the work just aims at obtaining results consistent with previous findings. Therefore, I would suggest to underline that the novelty stands in the approach adopted, revise the related work by examining previous similar approaches, and emphasize why it could be preferable (e.g., number of samples) but also which are its limits, with respect to the state of the art approaches (e.g., randomized trials).

Response

The key point in our paper is that using administrative data can generate comparable finding with randomized control trials. But administrative data are cheaper to obtain, easier to update and most importantly have much larger sample size (in fact, we can have the whole population). Thus, we demonstrate that using administrative data in disease progression modelling will be more efficient and have stronger external validity.  The novelty of our study is the ability to control for late enrolment bias and death rate bias, which we have discussed in Section 2.5.

Reviewer 2 Report

This study “ Can Administrative Health Data Improve the Gold Standard? Evidence from a Model the Progression of Myocardial Infarction” use a multistate model (hidden Markov model) to investigate the accuracy of reporting myocardial infarction cases. Overall, the idea is good and scientifically important but the paper is poorly written.

Comments and suggestions:

Title:

Line 3: Please, fix the title and make it more clear “.. Evidence from a Model the Progression…”

Abstract:

No introduction and adequate method description are present in the abstract. Thus, Results are poorly written. Please, rewrite the abstract. Reading scientific papers in good journals help with writing skills.

Introduction:

Line 30 - use more updates references. The phrase “..Although the incidence of myocardial infarction is declining among developed countries…” is not true.

Inset description of the hidden Markov model

Methods:

Line 102: Table has no title and need to accommodate better the columns names. This should be considered table 1.

References:

More updated and recent references support better the data and discussion. Please, add them.

Author Response

Reviewer 2

Line 3: Please, fix the title and make it more clear “.. Evidence from a Model the Progression…”

Response

We insert the missing “of” in the title and now it’s clear.

Abstract:

No introduction and adequate method description are present in the abstract. Thus, Results are poorly written. Please, rewrite the abstract. Reading scientific papers in good journals help with writing skills.

Response

Thanks for this comment. We have rewritten the abstract by adding an introduction and a better description of the methods used.

Introduction:

Line 30 - use more updates references. The phrase “..Although the incidence of myocardial infarction is declining among developed countries…” is not true.

Response

We agree. Some developed countries may experience an increasing trend of myocardial infarction despite we found evidence of declining in some developed countries in the literature. We have updated the reference and rephrased the argument as “Although the incidence of myocardial infarction is declining in some developed countries [3], Australia is an outlier with a very high incidence rate [4,5].

Inset description of the hidden Markov model

Response

We have added more description of the model in Section 2.4.

Methods:

Line 102: Table has no title and need to accommodate better the columns names. This should be considered table 1.

Response

This is the transition rate matrix; we have added a row name and column for the ease of reading but it may confuse readers as a table. We have reformatted the matrix and removed names to make it less confusing as a table.

References:

More updated and recent references support better the data and discussion. Please, add them.

Response

We have updated the relevant references in data and discussion sections [3, 21,26,].

Round 2

Reviewer 1 Report

Some of my previous comments were considered to review the paper.

However, I still have some concerns.

First, regarding probabilities in Table 1: A sentence states "There was 43.9% of patients experienced an NSTEMI, which resulted in 4.9% death." This would correspond to 2.2% death rate, while Table 1 reports 6% death rate for females and 5.3% for males. This should be fixed, and all reported probabilities should be explained in detail.

Moreover, as a reader, I read and understand that “administrative data are cheaper to obtain, easier to update and most importantly have much larger sample size”. But, since “The key point in our paper is that using administrative data can generate comparable finding with randomized control trials.”, as a reader, I would like to read if this key point was questionable, based on previous works if any, in terms of limits of the finding. In this case, the fact that “Thus, we demonstrate that using administrative data in disease progression modelling will be more efficient and have stronger external validity.” would be a nice finding. I would also emphasize that this has been reached thanks to your contribution: “The novelty of our study is the ability to control for late enrolment bias and death rate bias, which we have discussed in Section 2.5.”